# Rehabilitation cooperation and person-centred dialogue meeting for patients sick-listed for common mental disorders: 12 months follow-up of sick leave days, symptoms of depression, anxiety, stress and work ability – a pragmatic cluster randomised controlled trial from the CO-WORK-CARE project

Cecilia Björkelund [1], Ausra Saxvik,[1] Irene Svenningsson [1,2]
Eva-Lisa Petersson,[1,2] Lilian Wiegner,[1] Maria Larsson [3], Karin Törnbom,[1,4]
Carl Wikberg,[1] Nashmil Ariai,[1] Shabnam Nejati,[1] Gunnel Hensing,[5]
Dominique Hange [1]

For numbered affiliations see end of article.

**Correspondence to**
Dr Cecilia Björkelund;
cecilia.bjorkelund@allmed.gu.se

## ABSTRACT

**Objectives** To study whether early and enhanced cooperation within the primary care centres (PCC) combined with workplace cooperation via a person-centred employer dialogue meeting can reduce days on sick leave compared with usual care manager contact for patients on sick leave because of common mental disorders (CMD). Secondary aim: to study lapse of CMD symptoms, perceived Work Ability Index (WAI) and quality of life (QoL) during 12 months.

**Design** Pragmatic cluster randomised controlled trial, randomisation at PCC level.

**Setting** 28 PCCs in Region Västra Götaland, Sweden, with care manager organisation.

**Participants** 30 PCCs were invited, 28 (93%) accepted invitation (14 intervention, 14 control) and recruited 341 patients newly sick-listed because of CMD (n=185 at intervention, n=156 at control PCCs).

**Intervention** Complex intervention consisting of (1) early cooperation among general practitioner (GP), care manager and a rehabilitation coordinator, plus (2) a person-centred dialogue meeting between patient and employer within 3 months. Control group: regular contact with care manager.

**Main outcome measures** 12 months net and gross number of sick leave days at group level. Secondary outcomes: 12 months depression, anxiety, stress symptoms, perceived WAI and QoL (EuroQoL-5 Dimensional, EQ-5D).

**Results** No significant differences were found between intervention and control groups concerning days of sick leave (intervention net days of sick leave mean 102.48 (SE 13.76) vs control 96.29 (SE 12.38) p=0.73), return to work (HR 0.881, 95% CI 0.688 to 1.128), or CMD symptoms, WAI or EQ-5D after 12 months.

**Conclusions** It is not possible to speed up CMD patients' return to work or to reduce sick leave time by early and

## STRENGTHS AND LIMITATIONS OF THIS STUDY

⇒ The CO-WORK-CARE trial was accomplished entirely in primary care.
⇒ The intervention was adapted to a level possible to conduct in a Swedish primary care context with low level of resources for direct workplace intervention and cooperation.
⇒ The intervention model was person-centred and emphasised the individual's perceptions of workplace conditions facilitating own return to work.
⇒ COVID-19 could have had possible impact on the last months of follow-up for 12 months sick leave duration, however, this effect should have been comparable in the control group.
⇒ The complexity of the intervention based on the individual patient's preferences, needs and values, a structured cooperation among general practitioner, care manager and rehabilitation coordinator and a person-centred dialogue meeting with employer could be called into question concerning possible biasing factors.

enhanced coordination among GP, care manager and a rehabilitation coordinator, combined with early workplace contact over and above what 'usual' care manager contact during 3 months provides.

**Trial registration number** NCT03250026.

## BACKGROUND

The largest diagnostic group among sick-listed women and men in Sweden is common

mental disorders (CMD), that is, depression, anxiety syndromes and stress-related mental disorder (adjustment disorder, exhaustion disorder).[1] The yearly cumulative incidence is approximately 3% of the labour force.[1] Likewise, in other parts of Europe CMD are common among persons who are sick-listed, and work-related stress is increasingly identified as a contributing cause.[2-5] In fact, more than 50% of new sick leave cases in Sweden have adjustment disorder or exhaustion disorder (International Classification of Diseases (ICD) F43) as their certification diagnosis,[1 3] while depression disorder has decreased as certification diagnosis.[1] Many of these individuals return to work (RTW) after treatment, but for some RTW is slower. Knowledge of the most decisive factors for continuing sick leave is to a great extent lacking for CMD patients today,[6] but reasons include symptom severity and difficulties to adapt workplace or work tasks.[6 7] Long sick leave periods (>12 weeks) are common in Sweden, and most intervention studies have so far failed to show marked positive effects on RTW, except for interventions including increased contact with the workplace.[7 8] Prolonged RTW can lead to a more permanent exclusion from the labour market, and better methods are needed to prevent an excessively long sick leave period that could possibly be counterproductive.

Since the largest proportion of individuals with mental illness seek primary care, usually with symptoms that manifest in depressive and/or anxiety symptoms, primary healthcare centres are a relevant arena for interventions.[9] An organisation guided by collaborative care principles, in which a care manager coordinates care at the primary care centre (PCC) by maintaining a close and regular contact with patients and aligns efforts for their individual needs, has shown good results regarding mental and physical health improvements.[10-12] In a recent systematic review, a care manager organisation at the PCC was one of few factors found to increase RTW frequency when compared with already existing forms of organisation and actions.[12] Other outcomes related to team-based interventions at the PCC were increased quality of care, reduction of symptoms and increased Work Ability Index (WAI), function and quality of life (QoL).[10-14] A care manager organisation seems to enhance care approaches that allow complexity, person-centredness and interaction.[10 15] The collaborative care organisation handles person-centredness and complexity via the care manager, who also facilitates interaction with other psychosocial competencies, especially a psychologist/psychotherapist for the individual patient (in the Swedish study around 45%–50%[10]), but also a counsellor, physiotherapist and other contacts at the PCC when needed. However, beneficial effects have primarily been shown for primary care patients with depressive illness. Thus far, evidence is lacking both concerning the entire CMD diagnosis spectrum and especially for those with a symptom burden and WAI reduction high enough to necessitate sick leave.

In an earlier Swedish study, PRIM-CARE randomised controlled trial (RCT) (ClinicalTrials.gov Identifier:

NCT02378272), a care manager intervention at the PCC improved RTW and reduced sick leave for patients with depression compared with care as usual (CAU).[10] Consequently, a wide implementation of a care manager organisation at interested PCCs has been going on since 2015 in the Region Västra Götaland, Sweden. The present CO-WORK-CARE trial was initiated to examine whether a care manager contact combined with early structured enhanced cooperation among the care manager, rehabilitation coordinator and general practitioner (GP) at the PCC plus a person-centred dialogue meeting with the patient's employer would further increase and shorten RTW for CMD patients on sick leave compared with those with 'usual' care manager contact.

## Objectives

The primary aim was to study whether an early structured enhanced cooperation among the care manager, rehabilitation coordinator and GP combined with workplace cooperation via a person-centred dialogue meeting with the patient's employer can reduce sick leave time and increase RTW and WAI compared with care manager contact 'as usual'. A secondary aim was to study lapse of depression, anxiety and stress-related symptoms as well as perceived WAI and QoL during 12 months. Thus, we regarded it as important also to examine the lapse of mental symptoms to study the effects of the intervention on the patient's mental state. As the intervention was based on structured enhanced cooperation between several staff members within the PCC, randomisation at PCC level was chosen.

## METHODS
### Study setting and context

As described above, an implementation of the care manager function at interested PCCs has been ongoing since 2015 in the Region Västra Götaland, Sweden (inhabitants ~1.7 million; ~200 PCCs). The first part of the implementation was conducted as an RCT (ClinicalTrials.gov Identifier: NCT02378272).[10] Since the RCT showed positive effects concerning depression course, QoL and RTW,[10 14 15] a care manager (mostly a registered nurse) for patients with CMD including stress-related disorder (adjustment disorder+exhaustion disorder) was implemented at most PCCs in the region (at around 160 PCCs). The implementation of a care manager collaborative care organisation for a team-based care was also recommended for national implementation in the Swedish guidelines for care of depression and anxiety syndromes.[16]

In Sweden, primary care is organised in PCCs, which (besides several GPs) also have specially educated nurses, primary care psychologists, counsellors and physiotherapists, often working in team around the patient. The PCCs serve around 2000–30 000 listed patients (around 1 GP and 2 nurses per 2000 listed). All PCCs have, for example, a specially educated nurse who works part time with diabetes patient care together with the GP, preferably via

individually adjusted health promotion activities, while also guaranteeing accessibility and continuity for the diabetic patient. In a similar manner, the care manager's assignment is to work part time with CMD patients to guarantee accessibility and continuity for the CMD patient and promote psychopedagogical support to the patient during the illness. A care manager academic education, financed via the Region Västra Götaland county council, is mandatory (Care Manager for CMD, 7.5 higher education credits).

During the same time as the care manager organisation was implemented, an implementation of a function to coordinate the sick leave and rehabilitation process, that is, a rehabilitation coordinator, was also carried out at all PCCs in the region. A rehabilitation coordinator was initiated in 2016 via a national project with special governmental funding to the regions to offer coordinating work to patients on sick leave to promote RTW. A rehabilitation coordinator has an education in social insurance laws and regulations and vocational rehabilitation and serves as a regulation knowledge broker for the PCCs and a support for the patient's RTW process and workplace cooperation. The education standards vary between regions and organisations. In primary care, the rehabilitation coordinator often serves several PCCs in contrast to care managers who are part of the PCCs' ordinary personnel and working part of their duty as care managers. At some PCCs enrolled in this trial (11/28=39%; 5/14 intervention PCCs, 6/14 control PCCs) the care manager also served as rehabilitation coordinator.

### Trial design

The design was a pragmatic cluster RCT with complex intervention and thus randomisation was performed at PCC level (online supplemental table S1). The study adheres to Consolidated Standards of Reporting Trials (CONSORT) guidelines with extension to cluster randomised trials.

### Participants

Invitation to PCCs to participate in a RCT started in September 2017, and 30 PCCs in the Västra Götaland Region with an established organisation consisting of a care manager and affiliated rehabilitation coordinator were invited via the regional primary care R&D support organisation to take part in the CO-WORK-CARE study. Of the invited 30 PCCs, 28 (93%) PCCs accepted the invitation. All PCCs were informed about the trial both via information visits to every PCC before study start and by written confirmation which was obtained from every PCC, both intervention and control PCCs. All PCCs participated in educational activities concerning study protocol (figure 1, upper part). Thus, at all PCCs in the RCT, intervention as well as control, a care manager was working part-time at the PCC and a rehabilitation coordinator was affiliated.

### Randomisation and masking

Randomisation was conducted at PCC level. A total of 28 PCC were included and divided into two strata, rural

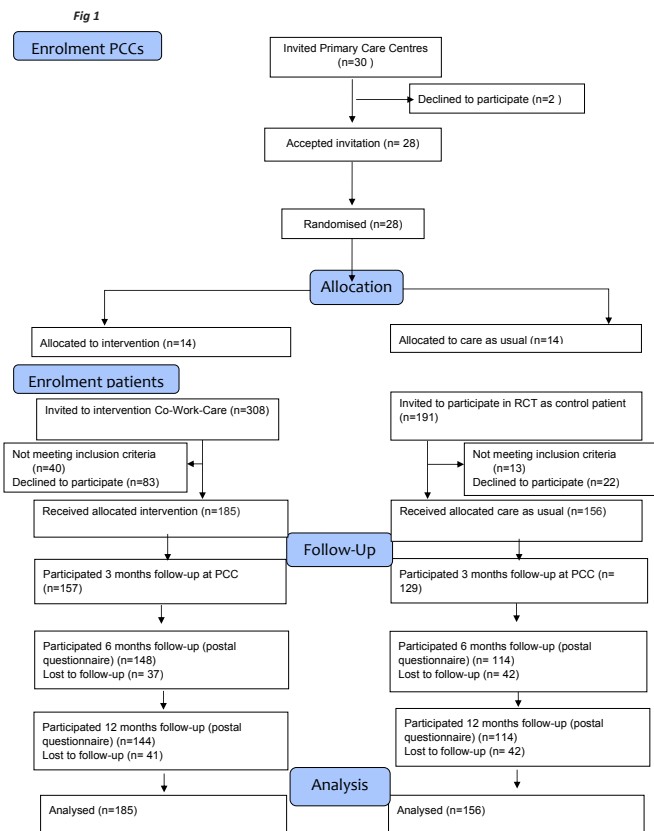

**Figure 1** Flow chart for the CO-WORK-CARE trial. Inclusion of primary care centres (PCCs) (upper part) and inclusion of patients, subdivided by intervention and control identification of primary care centres (lower part). RCT, randomised controlled trial.

(n=14) and urban (n=14). The centres in each stratum were then divided into seven blocks of two centres each. One centre in each block was randomised to the intervention group (n=14) and one to the control group (n=14) by the Primary Health Care R&D Region Västra Götaland, Sweden (figure 1). Cluster randomisation was chosen to avoid the treatment contamination between the intervention and control groups that would occur if patients at centres were individually randomised. No masking was possible, either concerning personnel at PCCs or research personnel.

### Patients

All newly attending patients at intervention and control PCCs who were judged in need of new sick leave due to mild/moderate depression (ICD F32, F33), anxiety syndrome (ICD F40, F41, F48) and/or stress-related mental disorder (ICD F43) (excluding post-traumatic stress disorder) were invited by the GP or the care manager to participate in the trial. Diagnosis was established by the GP, who made out the sickness certificate, and the diagnosis was confirmed by use of Primary Care Evaluation of Mental Disorders (PRIME-MD)[17], a diagnostic tool for depression and anxiety disorders especially for primary care, and by results from the Karolinska

**Table 1** Care Process in the Co-Work-Care trial

| Actor | GP | Care manager | Rehabilitation coordinator | Dialogue meeting |
|---|---|---|---|---|
| Action | Diagnoses CMD | Continuous contact from start sick leave certification | Initiates return to work | Within 12 weeks |
| First contact | Sick leave certification >14 days with CMD diagnosis. Somatic examination Information to care manager (and other staff when assessed needed) | Care plan together with patient based on interview and anamnesis. Assessment with MADRS-S GAD7 KEDS Information to other staff when therapy needed | Interview with patient when care manager indicates patient ready to start rehabilitation process. Time for dialogue meeting agreed on with patient. Dialogue meeting decided. | Person-centred dialogue meeting between patient and employer; rehabilitation coordinator as the dialogue moderator. Patient explains her/his view on return to work process |
| Follow-up | Evaluates need of further sick leave in regular reappointments | Telephone follow-up weeks 1, 2, 4, 8, 12 When patient ready: work questions discussed | Telephone contact with employer for interview about patient's work situation and information about dialogue meeting scope | |
| Ongoing | Structured cooperation between GP, care manager and rehabilitation coordinator | | | |

Grey parts common for control and intervention PCCs, green parts solely for intervention PCCs.
CMD, common mental disorder; GAD-7, General Anxiety Disorders Scale-7; GP, general practitioner; KEDS, Karolinska Exhaustion Disorder Scale; MADRS-S, Montgomery Asberg Depression Rating Scale-Self; PCC, primary care centres.

Exhaustion Disorder Scale (KEDS) assessment instrument (not diagnostic).[18][19]

### Exclusion criteria

Patients diagnosed with serious depression, bipolar disorder, psychosis, addiction, pregnancy > 1st trimester, cognitive impairment, post-traumatic stress syndrome or not speaking/understanding Swedish were excluded.

### Intervention

The CO-WORK-CARE complex intervention consisted of (1) early structured cooperation among care manager, rehabilitation coordinator and GP and (2) a person-centred dialogue meeting between patient and employer (immediate supervisor) with the rehabilitation coordinator as facilitator.

The care for the patient was based on the patient's ongoing sick leave, the sick leave diagnosis, the individual care plan (which the care manager and the patient initially drew up together) and the course of the CMD symptoms. In both intervention and control arms the individual care plan was drawn up between care manager and patient, but at intervention PCCs the rehabilitation coordinator was included from start, and the care manager, the rehabilitation coordinator and the GP continuously had contact about the lapse of the patient's illness and progress (see table 1). The intervention also consisted of a person-centred dialogue meeting between the employer and employee with the rehabilitation coordinator as the guide and facilitator, effectuated within 3 months from study start, as early as possible based on the patient's condition and opinion.

The purpose of the person-centred dialogue meeting was above all to make it possible for the patient to inform the employer about the patient's own conception of factors that could facilitate an RTW. The dialogue meeting was a simplified dialogue meeting applied to primary care context, adapted from a more extensive protocol by Karlson and Östberg.[20] The dialogue meeting was person-centred and focused on the needs of the individual patient and the patient's preferences, needs and values in the RTW process.[21] A person-centred dialogue meeting between the employer and the patient is not utilised in the Swedish workplace rehabilitation. Regulations state that a meeting among the workplace representative, the Social Insurance Agency, the PCC representative (most often GP, sometimes also rehabilitation coordinator) and the patient should be initiated after around 3 months of (full-time) sickness absence. This regulated meeting is not a person-centred meeting where the patient has the opportunity to describe her/his experience of the illness and thoughts about how to RTW at the workplace. In the CO-WORK-CARE study's person-centred dialogue meeting (where only three participants are present), the rehab coordinator was the mentor and guarantee for keeping the patient in the centre.

In all patient care, there was also a cooperation with the GP and with the PCC's psychosocial team, but the structured and early care manager/rehabilitation coordinator /GP cooperation as well as the person-centred dialogue meeting with the employer was the added value in the intervention.

### Procedures

#### Intervention PCCs

For the intervention PCCs, an extended instruction about the complex intervention with early cooperation

and person-centred dialogue meeting was conducted (8-hour session for the entire medical care personnel group including rehabilitation coordinator). During the RCT, each intervention PCC was regularly supported by a research team specialised nurse/counsellor to secure adherence to the complex intervention. The complex intervention with early cooperation and person-centred dialogue meeting was added to the care manager CAU function provided at the control PCCs, which already included engagement of psychologist/psychotherapist as well as other members of the psychosocial team according to national guidelines and focus on the patient's symptom development and preferences. Early cooperation was carried out by (1) information from the care manager to the rehabilitation coordinator about the patient's status and care planning concerning workplace contact in conjunction with the care manager's and patient's initial meeting and (2) in addition to CAU team meetings with the psychosocial team when psychologist/other staff were engaged, regular meetings among the care manager, GP and rehabilitation coordinator when the care manager indicated that the patient was ready to start rehabilitation process (table 1).

### Control PCCs

At the control PCCs, CAU consisted of: structured care manager function at the PCC with care manager contact with the patient during 3 months, as described in the PRIM-CARE trial.[10] The care manager function included an initial 1-hour meeting between patient and care manager, where a care plan was established adapted to the patient's condition and preferences, and an agreement about the communication between patient and care manager during the next 12 weeks, mainly contacts by telephone, was drawn up (table 1). The care manager contact was a way of increasing accessibility and continuity for the patient and was provided in addition to other therapeutic or pharmacological treatment according to the patient's needs.[10] This involved the engagement of a psychologist/ psychotherapist in around every second patient, as well as other members of the psychosocial team when needed.[10] Contacts with GP and other competencies were provided according to the needs of the individual patient. Contact with rehabilitation coordinator was often only made in case of long sick-list duration (usually >2 months).

Control PCCs were also supported concerning patient recruitment and follow-up.

### Patient and public involvement

Patients and/or the public were not involved in the design, conduct, or reporting, or dissemination plans of this research.

### Outcomes

Primary outcomes: Number of sick leave days (collected via the Micro-Data for Analysis of the Social Insurance System (database)) after 12 months at group level (intervention vs control patients) for comparison. Proportion of patients who returned to full time work, time until full RTW, number/proportion of patients on sick leave at 3, 6 and 12 months.

Secondary outcomes: Patient-related outcome measures (PROMs) with self-assessment instruments Montgomery Asberg Depression Rating Scale-Self (MADRS-S),[22] General Anxiety Disorders Scale-7 (GAD-7),[23] KEDS,[18] EuroQoL-5 Dimensional (EQ-5D)[24] and WAI[25] course. Remission according to MADRS-S (≤12), GAD (≤4), KEDS (≤18), frequency of antidepressant therapy, frequency of psychotherapeutic contacts.

### Self-assessment instruments used for PROMs

Depression was measured with the MADRS-S, which includes rating of mood, feelings of unease, sleep, appetite, ability to concentrate, initiative, emotional involvement, pessimism and zest for life.[22] Every item is rated from 0 to 6 points with a total of 54 points where points 0–12 are classified as no depression, 13–19 points as mild depression, 20–34 points as moderate depression and >34 points as severe depression.[22] We included individuals with ≤34 points.

Anxiety was measured by the GAD-7.[23] The GAD-7 consists of 7 items. Each question gives 0–3 points with a total of 21 points, classified as follows: >4 as mild GAD, >9 as moderate GAD and >14 as severe GAD.[23]

Exhaustion disorder was measured by the KEDS.[18] KEDS is a self-rating scale which consists of nine items, each ranging from 0 to 6 points with a scale range of 0–54, where points >18 indicate an increased risk for exhaustion disorder.[18]

To measure health-related QoL, the EQ-5D questionnaire was used, measuring health status with one question for each of the following five dimensions: mobility, self-care, usual activities, pain/discomfort and anxiety/ depression.[24]

Work ability was measured by the WAI.[25] WAI is a summary measure of seven items (range 7–49): current WAI, WAI in relation to the demands of the job, number of current diseases, estimated work impairment, sick leave during the past year, own prognosis of WAI and mental resources.[25]

### Statistics

Standard statistical methods were used for descriptive statistics. Continuous variables were analysed by independent sample t-test or Mann-Whitney U test and categorical variables and frequencies by Pearson $\chi^2$ test. Linear mixed model analysis with repeated measures was used to compare means of intraindividual change in depressive, anxiety and stress symptoms and QoL scores between the intervention group and the control group. These analyses were adjusted for cluster randomisation of PCC, repeated measures for every person and prespecified variables age, sex, education and antidepressants at inclusion. In the model were a repeated fixed effect for 'time' for each 'person' and 'centre' and a random intercept for each 'centre' included. These analyses to control for

correlation in repeated measures for each 'person' and 'centre' and to adjust for cluster randomisation of PCC. Linear mixed model analysis was performed for comparison of total number of 12 months sick leave days between intervention and control group and were adjusted for cluster randomisation of PCC, age, sex, education and antidepressants at inclusion. A random intercept for each 'centre' was included in the model adjusting for cluster randomisation of PCC. EMMEANS subcommand from Mixed model in SPSS were used to compare estimated marginal means between the intervention group and the control group for every time in the study. Cox regression analysis was done to study RTW between the intervention and the control group during 12 months of sick leave with control of the variables age, sex and education with no adjustment for cluster randomisation of PCC.

The statistical analyses were made using statistical software SPSS, V.28 and SAS, V.9.4. Statistical significance was set at p<0.05. No multiple adjustments were considered.

### Power calculation

The trial was designed to achieve adequate power for the primary outcome. Based on a recently accomplished RCT study[10] with randomisation at the PCC level, where a significant difference in RTW could be identified in a group of 83 vs 88 patients on sick leave at baseline in intervention and control groups, we made the assumption that 150 patients in each group would be sufficient for reaching a power of 80% to detect a difference of 3 units between the two groups with a significance level of 10%. The underlying assumption was a SD in the group of 10 units, a within-subject correlation of 0.4, and a within-cluster correlation of 0.1, that is, a design effect of 1.9 to correct for having a cluster analysis.

To be able to conduct analyses for the group of patients with stress-related mental disorder (adjustment disorder, exhaustion disorder), patients with F43 as main or contributing diagnosis were oversampled during 1 February 2020–30 June 2020.

### RESULTS

Inclusion of patients started December 2017 and was completed end of June 2020. In total 499 patients recently on sick leave (<7 weeks) because of depression (F32, F33), anxiety syndrome (F40, F41, F48) and/or stress-related mental disorder (adjustment disorder, exhaustion disorder) (F43) were invited (figure 1 flow chart). At the intervention PCCs, 308 patients were initially invited, but 40 did not meet inclusion criteria, 83 declined to participate and 185 agreed to participate (89%). At the control PCCs, 191 patients were initially invited, but 13 did not meet inclusion criteria, 22 declined to participate and 156 accepted to participate (88%). In total, 341 patients participated, 54% in intervention and 46% in control group.

Table 2 shows baseline data for intervention and control groups. There were differences concerning age (lower

mean age control group) and percentage of men (higher in control group) as well as sedentary level of leisure time physical activity (higher level sedentary in control group).

At the intervention PCCs, early cooperation among care managers, rehabilitation coordinators and the patients' GPs was carried out. Collaboration with team members such as psychotherapists, medical social workers, physiotherapists and occupational therapists was carried out when this was deemed relevant/needed. Out of the 185 intervention group patients, 102 (55%) had a person-centred dialogue meeting with the employer within 3 months, in most cases at the PCC. The reason for not having a dialogue meeting was mainly early RTW. There was a significant difference in mean number of days on sick leave between patients who had and patients who did not have a dialogue meeting: mean number of days on sick leave, 192 vs 117 (patients with vs patients with no dialogue meeting), p<0.001.

### Vocational outcomes: primary outcome

Analysis of gross and net number of days on sick leave for intervention and control groups in the CO-WORK-CARE trial for CMD diagnoses during 12 months follow-up showed that there was no significant difference concerning net number of sick leave days between intervention and control groups (table 3). The analyses were conducted both for sick leave due to CMD and for sick leave due to all diagnoses (mental as well as somatic) during 12 months. All these analyses showed non-significant results for sick leave days but with shorter duration for control group patients. We also analysed total number of sick leave days for patients with mild CMD only (n=57), as well as patients with moderate-high CMD symptoms (n=284), both of which analyses did not show any significant differences concerning sick leave days (table 3).

Online supplemental figure S1 shows survival plot from Cox regression analysis of cumulative return to work (full time or part time sick-leave to no sick-leave), gross number of days on sick-leave 12 months, controlled for age, gender and education. There was no significant difference between intervention and control groups, hazard ratio (HR)=0.881 with 95% CI 0.688 to 1.128.

### Secondary outcomes (PROMs)

There were no significant differences in levels at 3, 6 and 12 months between the intervention and control groups concerning depression (MADRS-S), anxiety (GAD-7) or stress (KEDS) symptoms, WAI or QoL (EQ-5D) (table 4, online supplemental figure S2). For participants with medium/high levels of depression (MADRS-S) and anxiety (GAD-7) symptoms, there were significant differences at 12 months between intervention and control groups, with lower levels in the intervention group. However, there were no differences at all concerning stress symptoms (KEDS) (table 4, online supplemental figure S3).

Table 5 shows participants on sick leave at baseline, 3, 6 and 12 months. At 3 months, there was a significant

**Table 2** Baseline data for intervention (n=185) and control group (n=156) patients, total=341 patients, included in the CO-WORK-CARE trial (December 2017 until June 2020)

| | Total patient group | | Intervention group | | Control group | |
|---|---|---|---|---|---|---|
| | n | % | n | % | n | % |
| Women | 275 | 80.6 | 157 | 84.9 | 118 | 75.6 |
| Men | 66 | 19.4 | 28 | 15.1 | 38 | 24.4 |
| Occupation Working | 328 | 96.8 | 178 | 96.2 | 150 | 97.4 |
| Studying | 1 | 0.3 | 0 | 0 | 1 | 0.6 |
| In search of work/other | 10 | 2.9 | 7 | 3.8 | 3 | 1.9 |
| Hours of work Full time | 299 | 87.9 | 164 | 88.6 | 135 | 87.1 |
| Other (25%–75%) | 41 | 12.1 | 21 | 11.4 | 20 | 12.9 |
| Marital status Cohabiting | 250 | 74.0 | 142 | 78 | 108 | 69.2 |
| Single | 88 | 26.0 | 40 | 22 | 48 | 30.8 |
| Born Outside Nordic Country | 32 | 9.4 | 14 | 7.6 | 18 | 11.5 |
| Educational level Primary | 20 | 5.9 | 13 | 7 | 7 | 4.5 |
| Secondary | 194 | 56.9 | 105 | 56.8 | 89 | 57.0 |
| University/college | 127 | 37.2 | 67 | 36.2 | 60 | 38.5 |
| Physical activity leisure time sedentary | 55 | 16.3 | 23 | 12.6 | 32 | 20.8 |
| Smoking yes+sometimes | 77 | 22.6 | 36 | 19.5 | 41 | 26.5 |
| Alcohol high (>8 p AUDIT) | 27 | 9.0 | 16 | 9.8 | 11 | 8.1 |
| Low socioeconomic status | 178 | 58.6 | 96 | 57.1 | 82 | 60.3 |
| Antidepressant medication | 110 | 32.3 | 63 | 34.1 | 47 | 30.1 |
| Sick leave last year, self-reported (yes) | 132 | 40.9 | 78 | 43.8 | 54 | 37.2 |
| Sick leave diagnosis | | | | | | |
| Depression (F32, F33) | 64 | 18.8 | 39 | 21.1 | 25 | 16 |
| Anxiety syndrome (F40, F41, F48) | 48 | 14.1 | 22 | 11.9 | 26 | 16.7 |
| Stress related mental disorder (F43) | 200 | 58.7 | 111 | 60.0 | 89 | 57.1 |
| | Mean | SD | Mean | SD | Mean | SD |
| Age | 41.3 | 11.2 | 44.2 | 10.9 | 37.9 | 10.7 |
| MADRS-S | 22.3 | 8.0 | 21.7 | 8.0 | 22.9 | 8.1 |
| GAD-7 | 11.6 | 4.9 | 11.2 | 4.7 | 12.2 | 5.1 |
| KEDS | 28.7 | 8.7 | 28.2 | 8.5 | 29.3 | 9.1 |
| EQ-5D | 42.8 | 17.3 | 43.8 | 17.1 | 41.5 | 17.6 |
| WAI | 2.56 | 2.4 | 2.43 | 2.4 | 2.71 | 2.4 |

EQ-5D, EuroQoL-5 Dimensional; GAD-7, General Anxiety Disorders Scale-7; KEDS, Karolinska Exhaustion Disorder Scale; MADRS-S, Montgomery Asberg Depression Rating Scale-Self; WAI, Work Ability Index.

difference between intervention and control groups with lower percentage of patients on sick leave in the control group, but no significant differences at 6 and 12 months. Table 5 also shows comparisons between participants in intervention and control groups with mild and moderate-to-high levels, respectively, concerning depression symptoms, anxiety symptoms and stress symptoms at respective point in time.

Percentage of patients on antidepressants and patients receiving psychotherapy contact did not show any significant differences at 3, 6 or 12 months (table 5).

### Subgroup analyses

Analysis including only patients that participated in a person-centred dialogue meeting (n=102) compared with control group did not reveal different results from comparison of the total intervention and control groups.

### Harmful events

No harmful events, suicide attempts or psychiatric hospital care were registered for patients included in the CO-WORK-CARE trial during the 12 months observation time.

**Table 3** Gross and net number of days on sick leave for intervention and control groups in the CO-WORK-CARE trial for CMD diagnose (depression and anxiety disorders+adjustment and exhaustion disorders), for all diagnoses and for net number of days for mild and moderate-high CMD diagnoses, respectively

| | Intervention | | Control | | |
| --- | --- | --- | --- | --- | --- |
| | Mean | SE | Mean | SE | P value |
| GROSS total no of days CMD diagnoses | 147.30 | 16.96 | 125.96 | 15.16 | 0.331 |
| NET total no of days CMD diagnoses | 102.48 | 13.76 | 96.29 | 12.38 | 0.726 |
| GROSS total no of days all diagnoses | 153.66 | 16.24 | 135.82 | 14.38 | 0.391 |
| NET total no of days all diagnoses | 108.98 | 13.08 | 104.99 | 11.68 | 0.810 |
| NET total no of days mild CMD diagnoses (n=57) | 82.44 | 20.55 | 80.97 | 21.05 | 0.959 |
| NET total no of days moderate-high CMD diagnoses (n=284) | 115.12 | 14.12 | 110.69 | 12.63 | 0.803 |

Mixed model analysis adjusted for clustering and age, sex, education and antidepressants at inclusion; comparison of total number of 12 months sick leave days between intervention and control group.
CMD, common mental disorder.

## DISCUSSION
### Main findings
The main finding was that there were no differences in sick leave days between intervention and control groups. A structured early cooperation among care manager, GP and rehabilitation coordinator at the PCC combined with a person-centred dialogue meeting between the sick-listed person and employer, with the rehabilitation coordinator as a moderator, did not lead to shortened sick leave time in comparison with usual care manager contact only. Patients with CMD diagnoses—both when sick leave days were studied for CMD diagnoses only as well as total sick leave for all diagnoses—showed no significant differences concerning sick leave days.

However, for patients with moderate-to-high symptom levels of depression and anxiety, there were also significant differences in levels of CMD depression and/or anxiety symptoms at 12 months between intervention and control groups, but no difference at all concerning stress symptoms. This indicates that a closer cooperation and a planned person-centred dialogue meeting could be beneficial for the mental well-being of persons with moderate-to-high symptom levels of depression/anxiety, as indicated by significantly lower levels of depression and anxiety symptoms after 12 months. This is an important finding, as individuals with severe symptoms of depression and anxiety may be the most difficult to help and have been shown to have the longest sick leave period among patients with CMD.[6 7]

### Strengths and weaknesses
The strengths of this study are several. The trial was accomplished entirely in primary care. All participating PCC personnel as well as research personnel had very good contextual knowledge, and the intervention was adapted to a level possible to conduct in a Swedish primary care context with low level of resources for direct workplace intervention and cooperation. The intervention model was person-centred and emphasised the individual's perceptions of workplace conditions facilitating own

RTW. We succeeded in achieving the planned sample size of patients, despite the occurrence of the COVID-19 epidemic during the last 3 months before the 12 months follow-up was finalised for the latest included participants.

Among limitations, the risk of identification and recruitment bias was a limitation since providers were aware of being in the intervention or control arm. More patients participated in the intervention arm than in the control arm, and there were some differences between the arms in the baseline characteristics of recruited patients. The impact that COVID-19 could have had on the last months of follow-up for RTW and 12 months sick leave duration should also be mentioned. However, this effect should have been comparable in the control group. Another limitation was that only around 55% of patients in the intervention group took part in the person-centred dialogue meeting. A major reason for this was that the dialogue meeting was not deemed relevant or needed for the patients with early RTW in the fast rehabilitation process, and this was mostly due to the patient's own preference. Another limitation was that multiplicity was not accounted for in the analyses of secondary outcomes.

The complexity of the intervention based on the individual patient's preferences, needs and values, a structured cooperation among GP, care manager and rehabilitation coordinator and a person-centred dialogue meeting with employer could be called into question concerning possible biasing factors. However, primary care is a complex clinical context where the handling of care problems is guided by the individual patient's needs as the starting point, and the pragmatic design of the trial compensates at least partly for multiple biasing factors.

A further limitation was that the trial was only possible to carry out in a primary healthcare system where a collaborative care model and team-based care were implemented. However, the collaborative care model with a care manager function has also been implemented in several European countries such as the UK, Netherlands and Italy, in addition to Canada and the USA.[26] The care manager organisation

**Table 4** Start to 3, 6 and 12 months follow-up of differences of mean values between intervention and control group of MADRS-S, GAD-7, KEDS, EQ-5D and WAI in the CO-WORK-CARE trial, all patients and patients with low and moderate high levels, respectively

| | Intervention mean (SE) | Control mean (SE) | Mean difference (SE) | 95% CI | P value |
|---|---|---|---|---|---|
| MADRS-S at start | 22.14 (0.81) | 23.04 (0.81) | −0.896 (1.10) | −3.126, 1.335 | 0.421 |
| MADRS-S 3 months | 14.01 (0.86) | 15.18 (0.88) | −1.17 (1.18) | −3.549, 1.199 | 0.326 |
| MADRS-S 6 months | 12.51 (0.86) | 13.61 (0.90) | −1.10 (1.19) | −3.490, 1.292 | 0.361 |
| MADRS-S 12 months | 11.03 (0.87) | 12.79 (0.89) | −1.76 (1.19) | −4.158, 0.631 | 0.146 |
| GAD-7 at start | 11.35 (0.44) | 11.87 (0.45) | −0.524 (0.597) | −1.712, 0.664 | 0.383 |
| GAD-7 3 months | 6.51 (0.47) | 7.61 (0.50) | −1.10 (0.66) | −2.405, 0.200 | 0.096 |
| GAD-7 6 months | 5.85 (0.43) | 6.37 (0.46) | −0.52 (0.59) | −1.689, 0.658 | 0.385 |
| GAD-7 12 months | 5.49 (0.46) | 6.62 (0.49) | −1.13 (0.64) | −2.402, 0.140 | 0.081 |
| KEDS at start | 28.11 (1.09) | 28.93 (1.06) | −0.823 (1.47) | −3.814, 2.167 | 0.579 |
| KEDS 3 months | 19.43 (1.15) | 20.52 (1.16) | −1.082 (1.59) | −4.275, 2.110 | 0.498 |
| KEDS 6 months | 17.54 (1.17) | 18.22 (1.18) | −0.679 (1.62) | −3.926, 2.568 | 0.676 |
| KEDS 12 months | 15.69 (1.19) | 16.78 (1.20) | −1.095 (1.65) | −4.399, 2.209 | 0.509 |
| EQ-5D at start | 43.83 (1.89) | 40.42 (1.86) | 3.41 (2.54) | −1.748, 8.569 | 0.188 |
| EQ-5D 3 months | 63.02 (2.10) | 60.29 (2.15) | 2.73 (2.91) | −3.086, 8.552 | 0.351 |
| EQ-5D 6 months | 71.13 (2.08) | 66.23 (2.15) | 4.90 (2.88) | −0.866, 10.669 | 0.094 |
| EQ-5D 12 months | 72.34 (2.08) | 67.92 (2.12) | 4.43 (2.87) | −1.309, 10.168 | 0.128 |
| WAI at start | 2.53 (0.29) | 2.65 (0.28) | −0.12 (0.39) | −0.901, 0.663 | 0.761 |
| WAI 3 months | 5.48 (0.32) | 5.42 (0.33) | 0.06 (0.44) | −0.833, 0.944 | 0.901 |
| WAI 6 months | 6.94 (0.31) | 6.45 (0.31) | 0.49 (0.43) | −0.360, 1.340 | 0.254 |
| WAI 12 months | 7.55 (0.31) | 7.08 (0.31) | 0.47 (0.43) | −0.376, 1.323 | 0.269 |
| MADRS-S>19 at start | 26.97 (0.63) | 28.19 (0.62) | −1.21 (0.82) | −2.880, 0.457 | 0.150 |
| MADRS-S>19, 3 months | 16.16 (0.91) | 18.44 (0.96) | −2.28 (1.28) | −4.822, 0.255 | 0.078 |
| MADRS-S>19, 6 months | 14.04 (0.96) | 16.35 (1.05) | −2.31 (1.37) | −5.016, 0.396 | 0.094 |
| MADRS-S>19, 12 months | 11.71 (0.92) | 15.30 (1.30) | −3.59 (1.31) | −6.189, −0.993 | **0.007** |
| GAD-7>9 at start | 14.03 (0.37) | 15.13 (0.37) | −1.10 (0.597) | −2.089, −0.112 | **0.030** |
| GAD-7>9, 3 months | 7.32 (0.58) | 8.73 (0.63) | −1.42 (0.83) | −3.069, 0.230 | 0.091 |
| GAD-7>9, 6 months | 6.39 (0.53) | 7.80 (0.58) | −1.41 (0.76) | −2.908, 0.094 | 0.066 |
| GAD-7>9, 12 months | 5.54 (0.58) | 7.89 (0.63) | −2.34 (0.83) | −3.985, −0.696 | **0.006** |
| KEDS>29 at start | 34.88 (0.73) | 36.34 (0.70) | −1.46 (0.93) | −3.365, 0.445 | 0.128 |
| KEDS>29, 3 months | 23.46 (1.14) | 25.00 (1.19) | −1.537 (1.60) | −4.704, 1.631 | 0.339 |
| KEDS>29, 6 months | 21.25 (1.23) | 22.27 (1.28) | −1.02 (1.73) | −4.436, 2.404 | 0.558 |
| KEDS>29, 12 months | 18.56 (1.23) | 20.55 (1.30) | −1.99 (1.73) | −5.428, 1.447 | 0.254 |

Bold figures, p<0.05.
Mixed model analysis adjusted for clustering, age, sex, education and antidepressants at inclusion.
CMD, common mental disorder; EQ-5D, EuroQoL-5 Dimensional; GAD-7, General Anxiety Disorders Scale-7; KEDS, Karolinska Exhaustion Disorder Scale; MADRS-S, Montgomery Asberg Depression Rating Scale-Self; WAI, Work Ability Index.

implemented in the Region Västra Götaland has been judged to represent a collaborative care programme for depression and anxiety disorder that makes possible personalised care planning and shared decision-making.[27]

### Other studies
Several complex interventions, of which some mainly in the primary care context, aiming at increasing RTW and reducing sick leave time have been performed. Bakker *et al* already published a cluster RCT consisting of a minimal intervention for patients with stress-related mental disorders on sick leave in primary care (MISS) that aimed at reducing sick leave and preventing chronicity of symptoms.[3] No superior effect of MISS was found concerning duration of sick leave or severity of symptoms compared

**Table 5** Number and percentage of participants in intervention and control groups with mild and moderate-to-high levels, respectively, of depression symptoms, anxiety symptoms and stress symptoms according to MADRS-S, GAD-7 and KEDS

| | Baseline | | 3 months | | 6 months | | 12 months | |
|---|---|---|---|---|---|---|---|---|
| | Intervention | Control | Intervention | Control | Intervention | Control | Intervention | Control |
| | n (%) | n (%) | n (%) | n (%) | n (%) | n (%) | n (%) | n (%) |
| On sick leave | 184 (99.5) | 156 (100) | **114 (62.0)** | **80 (51.3)** | 66 (35.9) | 46 (29.5) | 28 (15.2) | 24 (15.4) |
| MADRS-S>12 | 160 (87.4) | 143 (91.7) | **72 (46.5)** | **75 (58.6)** | 62 (42.5) | 54 (48.6) | 49 (34.3) | 49 (43.4) |
| MADRS-S>19 | 116 (63.4) | 100 (64.1) | 39 (25.2) | 35 (27.3) | 27 (18.5) | 25 (22.5) | **15 (10.5)** | **23 (20.4)** |
| GAD>4 | 165 (92.2) | 140 (92.1) | 91 (59.9) | 90 (70.9) | 87 (59.6)' | 71 (63.4) | 73 (51.8) | 61 (53.5) |
| GAD>9 | 120 (67.0) | 99 (65.1) | 33 (21.7) | 40 (31.5) | **21 (14.4)** | **28 (25.0)** | **19 (13.5)** | **31 (27.2)** |
| KEDS>18 | 162 (88.5) | 132 (86.5) | 82 (52.2) | 77 (60.2) | 68 (46.6) | 52 (46.0) | 52 (36.4) | 44 (38.9) |
| KEDS>29 | 94 (51.4) | 83 (54.6) | 26 (16.6) | 27 (21.1) | 17 (11.6) | 18 (15.9) | 11 (7.7) | 12 (10.6) |
| Anti-depressant use | 63 (34.1) | 47 (30.1) | 60 (38.2) | 50 (39.1) | 54 (36.5) | 41 (36.0) | 49 (34.0) | 40 (35.1) |
| Psychotherapy contact | – | – | 70 (44.6) | 71 (55.0) | 58 (39.2) | 39 (34.2) | 28 (19.4) | 15 (13.2) |

Levels at baseline, 3, 6 and 12 months, as well as number and percentage of participants on sick leave at respective point in time, on antidepressants and receiving psychotherapy contact.
Bold figures indicate significant difference between intervention and control groups.
GAD, General Anxiety Disorders; KEDS, Karolinska Exhaustion Disorder Scale; MADRS-S, Montgomery Asberg Depression Rating Scale-Self.

with CAU (GP contact at PCCs) after 12 months.[3] The latest published RCT on interventions aimed at integrating vocational rehabilitation and mental healthcare in people with depression and anxiety, where an integrated intervention both in healthcare and at the workplace was compared with improved mental care and CAU, showed no increased RTW, although a higher proportion in work at 12 months was shown for the integrated intervention.[28] The authors concluded that the goal of a very fast RTW is not necessarily beneficial, and that integration gave slightly better outcomes, but the clinical significance, that is, health advantage, was questionable.[28]

### Interpretation of findings
The results of the trial indicate that providing more intense contacts with care and the workplace for patients with CMD, with a symptom burden and reduction in WAI high enough to cause need of sick leave, did not lead to earlier return to full time work participation compared with the usual level of support to the patient provided by the care manager in a collaborative care organisation together with access to psychological and other therapeutic competences at the PCC. It seems that it is not possible to hurry up the processes that these patients need. Building a person-centred care process based on accessibility, continuity and close contact with a care manager makes it possible to adjust the care and timing to the individual patent's needs. The level of support and care that the care manager provides for the patient during the initial 3 months of the CMD illness seems 'good enough'. Extra care efforts on top of this, as provided in this intervention, do not seem to improve the patient's state. It is possible that a more thorough dialogue with greater competency is needed or, alternatively, repeated meetings to achieve a faster RTW. The person-centred

dialogue meeting may achieve a better understanding on the part of the employer for the worker's health situation but a change in the work situation might not be possible anyway.

RCT trials have shown effectiveness of collaborative care and care management in patients with depression, and the CO-WORK-CARE trial further develops the collaborative care/care manager intervention with structured early cooperation between the PCC competencies, including the GP and rehabilitation personnel, and patients on sick leave because of depression, anxiety and stress-related mental disorder. Care management was the basis of care in both arms, while the early and structured cooperation among the care manager, rehabilitation coordinator and the GP coupled with the person-centred dialogue meeting with employer the was the added value of the intervention. The effects of the structured cooperation were only discernible on the patient level. Early and structured cooperation coupled with a dialogue meeting can thus be said to be more effective in the long run for patients concerning their moderate-serious depression and anxiety symptoms than usual care manager contact, but not at all more effective concerning sick leave duration. Interview studies of patients' and personnel's perceptions of working according to the CO-WORK-CARE model reveal that patients perceived the person-centred dialogue meeting with the employer as positive,[29] and care managers and rehabilitation coordinators appreciated the cooperation.[30] The doctors in particular regarded the CO-WORK-CARE model as one that facilitated their often complex and frustrating work situation, and they perceived that they gained better control and support for the care of the patient through the close collaboration.[31]

## Implications

A good enough way to help the patient back to work in a salutogenic way seems primarily to be to provide accessibility, continuity, close contact and emphasis on person-centred care, to follow the patient continuously and to await start of workplace contact until the patient is ready for this.

### Author affiliations
¹Primary Health Care/School of Public Health and Community Medicine/Sahlgrenska Academy, University of Gothenburg, Gothenburg, Sweden
²Research and Development Primary Health Care, Region Västra Götaland, Gothenburg, Sweden
³Section for rehabilitation and Health, Institute of Neuroscience and Physiology, Sahlgrenska Academy, University of Gothenburg, Goteborg, Sweden
⁴Department of Social Work, University of Gothenburg, Goteborg, Sweden
⁵School of Public Health and Community Medicine, Sahlgrenska Academy, University of Gothenburg, Gothenburg, Sweden

**Acknowledgements** We want to thank all primary care centres in Västra Götaland who have been engaged in the CO-WORK-CARE study for their engagement and participation.

**Contributors** CB was the principal investigator and guarantor of the project. CB, E-LP, IS, DH and GH participated in the design of the work. CB, E-LP, IS, DH and NA handled the data and had the main responsibility for conducting the analyses, GH, SN contributed to conduction of additional analyses. CB, AS, GH, E-LP, IS, DH, NA, SN, KT, ML, CW and LW were responsible for the writing of the paper. All authors contributed to the interpretation of data and critical text revision and read and approved the final version of this manuscript. The corresponding author attests that all listed authors meet authorship criteria and that no others meeting the criteria have been omitted.

**Funding** This work was supported by Grants from Forte Dnr 2016-07412, Dnr 2018-01266 and grants from the Swedish state under the agreement between the Swedish government and the county councils, the ALF agreement (ALFGBG-722441, ALFGBG-965520).

**Disclaimer** The funding organisations have no role in the planning, execution or analyses of the trial.

**Competing interests** None declared.

**Patient and public involvement** Patients and/or the public were not involved in the design, or conduct, or reporting, or dissemination plans of this research.

**Patient consent for publication** Not applicable.

**Ethics approval** This study involves human participants and was approved by Regional Ethical Review Board Gothenburg, Sweden Dnr: 459-17; T745-18. Participants gave informed consent to participate in the study before taking part.

**Provenance and peer review** Not commissioned; externally peer reviewed.

**Data availability statement** Data are available on reasonable request. Data are not publicly available due to Swedish law but are available from the authors on reasonable request. The data are stored at the Gothenburg University, Arvid Wallgrens backe 7, 40530 Göteborg. Contact details: Cecilia Björkelund Orcid iDs: 0000-0003-4083-7342.

**ORCID iDs**
Cecilia Björkelund http://orcid.org/0000-0003-4083-7342
Irene Svenningsson http://orcid.org/0000-0002-7421-8171
Maria Larsson http://orcid.org/0000-0002-1225-9736
Dominique Hange http://orcid.org/0000-0003-1114-4440

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
