## [Reviewer comments · BMJ Open]

ARTICLE DETAILS

TITLE (PROVISIONAL)	Rehabilitation cooperation and person-centred dialogue meeting for patients sick-listed for common mental disorders - 12 months follow-up of sick-leave days, symptoms of depression, anxiety, stress and work ability. A pragmatic cluster-randomised controlled trial from the CO-WORK-CARE project
AUTHORS	Björkelund, Cecilia; Saxvik, Ausra; Svenningsson, Irene; Petersson, Eva-Lisa; Wiegner, Lilian; Larsson, Maria; Törnbohm, Karin; Wikberg, Carl; Ariai, Nashmil; Nejati, Shabnam; Hensing, Gunnel; Hange, Dominique

VERSION 1 – REVIEW

REVIEWER	Taljaard, Monica Ottawa Hospital Research Institute, Clinical Epidemiology Program
REVIEW RETURNED	24-Apr-2023

GENERAL COMMENTS	Thank you for the responses to all my previous queries. I unfortunately still have some concerns about the methodology that was used. 1. The issue of identification and recruitment bias cannot so easily be dismissed. It should at least be acknowledged as a limitation. Since providers were fully aware of being in the intervention or control arm, there is a substantial risk that different kinds of participants were recruited in the intervention and control arms. As I noted, a sign of this potential risk of bias is that many more patients were invited in the intervention arm than in the control arm. As you note, there were differences between the arms in the baseline characteristics of recruited patients. I encourage you to recognize this as a major limitation.2. The concern about adjustment for stratification factors in the analysis was not addressed. You stratified the centers according to urban or rural status. The recommended method of analysis is to adjust for urban/rural status as a cluster-level covariate in the analysis to obtain correct inferences. It is possible to include fixed cluster-level covariates (fixed effects) while accounting for cluster as a random effect. If this was not done, an explanation could be added to the manuscript.3. It is great that you managed to account for clustering in your analysis, although the description of the statistical analysis is very rudimentary and not clear. For example, the Cox regression analyses results did not change which suggests that you did not account for clustering in these analyses. Some additional clarification about how you accounted for the clustering in your other outcome analyses is also required. For example, you state that “mixed model analysis with repeated measures was used to
--

	compare means of intra-individual change in depressive, anxiety and stress symptoms and QoL scores between the intervention group and the control group". I assume this was linear mixed modeling for the secondary outcomes. Did this model account for correlation in repeated measures on the same individual as well as clustering by center? How was this done? How were comparisons between the arms obtained from the model? You also state that "GLM univariate analyses were performed for comparison of total number of sick-leave days between intervention and control group and were adjusted for clustering ...". How was this done? Do you mean that a random intercept was included for the cluster? What distribution was assumed – Poisson? Normal? Negative binomial? How were estimates of the intervention effect obtained? Perhaps some published examples with complete descriptions of statistical analysis methods from other published cluster randomized trials might be helpful. 4. Further clarification regarding your power calculation can be added to the manuscript. For example, if the trial was not designed to achieve adequate power for the primary outcome, this should be clearly stated. The sample size is typically driven by the primary trial objective (primary trial outcome). 5. You refer to some statistically significant results in the subgroup analyses, but there is a large number of statistical tests and it is not clear that multiplicity was accounted for in all the between-arm comparison (at start, 3 months, 6 months, 12 months etc). If multiplicity was not accounted for, the risk of spurious findings due to multiplicity could be acknowledged as a limitation.
--	---

REVIEWER	Brekke, Mette University of Oslo, Institute of Health and Society
REVIEW RETURNED	26-Apr-2023

GENERAL COMMENTS	You have answered all my questions and responded to my remarks in a satisfactory way. The only - very minor - thing I still see as unclear is: Under Methods, study setting: The 'PCCs are publicly or privately driven..' What does 'privately driven' mean? By whom? Certainly publicly funded, not privately?
--

VERSION 1 – AUTHOR RESPONSE

Reviewer: 1

Dr. Monica Taljaard, Ottawa Hospital Research Institute Comments to the Author:

Thank you for the responses to all my previous queries. I unfortunately still have some concerns about the methodology that was used.

1. The issue of identification and recruitment bias cannot so easily be dismissed. It should at least be acknowledged as a limitation. Since providers were fully aware of being in the intervention or control arm, there is a substantial risk that different kinds of participants were recruited in the intervention and control arms. As I noted, a sign of this potential risk of bias is that many more patients were invited in the intervention arm than in the control arm. As you note, there were differences between the arms in the baseline characteristics of recruited patients. I encourage you to recognize this as a major limitation.

ANSWER: Thank you for pointing this out! We have recognized this as a major limitation and added: the risk of identification and recruitment bias was a limitation since providers were aware of being in the intervention or control arm. More patients participated in the intervention arm than in the control

arm, and there were some differences between the arms in the baseline characteristics of recruited patients (page 26).

2. The concern about adjustment for stratification factors in the analysis was not addressed. You stratified the centers according to urban or rural status. The recommended method of analysis is to adjust for urban/rural status as a cluster-level covariate in the analysis to obtain correct inferences. It is possible to include fixed cluster-level covariates (fixed effects) while accounting for cluster as a random effect. If this was not done, an explanation could be added to the manuscript.

Answer: We have added to the manuscript under Statistical analysis.

3. It is great that you managed to account for clustering in your analysis, although the description of the statistical analysis is very rudimentary and not clear. For example, the Cox regression analyses results did not change which suggests that you did not account for clustering in these analyses. Some additional clarification about how you accounted for the clustering in your other outcome analyses is also required. For example, you state that "mixed model analysis with repeated measures was used to compare means of intra-individual change in depressive, anxiety and stress symptoms and QoL scores between the intervention group and the control group". I assume this was linear mixed modeling for the secondary outcomes. Did this model account for correlation in repeated measures on the same individual as well as clustering by center?

ANSWER: Answer: Yes exactly. We conducted a mixed model analysis using the MIXED command in SPSS to investigate the relationship between the dependent variable (for example "MADRS-S") and several independent variables, including the grouping variable (Intervention/control: "gr"), and the factors sex, education, and antidepressants at inclusion and Time, and Age. We used type III sum of squares and REML method to estimate parameters, and included a random intercept for each PCC and used a VC covariance structure. We also included a repeated effect for "Time" and used a diagonal covariance structure for each individual and PCC. We computed emmeans for "gr" and "Time" and tested for the interaction effect between "gr" and "Time".

.....
How was this done? How were comparisons between the arms obtained from the model? You also state that "GLM univariate analyses were performed for comparison of total number of sick-leave days between intervention and control group and were adjusted for clustering ...". How was this done? Do you mean that a random intercept was included for the cluster? What distribution was assumed – Poisson? Normal? Negative binomial? How were estimates of the intervention effect obtained?
.....

ANSWER: Sorry this is wrong, we forgot to correct in the text last time. It is linear mixed model analysis that is used here and not GLM univariate analyses. We performed a linear mixed model analysis using the MIXED command in SPSS to study the association between the dependent variable (sick-leave days) and several independent variables, including the grouping variable (Intervention/control:"gr"), and the factors sex, education, and antidepressants at inclusion and Time, and Age. We used type III sum of squares and REML method to estimate parameters, and printed the solution for further analysis. We included a random intercept for each PPC and used a VC covariance structure. We also included a repeated effect for Time and used a diagonal covariance structure for each individual and PCC. We computed emmeans for "gr" and "Time" and compared means for "gr" using the LSD test.

Perhaps some published examples with complete descriptions of statistical analysis methods from other published cluster randomized trials might be helpful.

Concluding ANSWER: We have expanded the statistical analysis part and added: In the model were a repeated fixed effect for "time" for each "person" and "center" and a random intercept for each "center" included. These analyses to control for correlation in repeated measures

for each "person" and "center" and to adjust for cluster randomisation of PCC. Linear mixed model analysis was performed for comparison of total number of 12 months sick-leave days between intervention and control group and were adjusted for cluster randomisation of PCC, age, sex, education and antidepressants at inclusion. A random intercept for each "center" was included in the model adjusting for cluster randomisation of PCC. EMMEANS subcommand from Mixed model in SPSS were used to compare estimated marginal means between the intervention group and the control group for every time in the study. Cox regression analysis was done to study RTW between the intervention and the control group during 12 months of sick-leave with control of the variables age, sex and education with no adjustment for cluster randomisation of PCC.

4. Further clarification regarding your power calculation can be added to the manuscript. For example, if the trial was not designed to achieve adequate power for the primary outcome, this should be clearly stated. The sample size is typically driven by the primary trial objective (primary trial outcome).

ANSWER: We have added under **Power analysis**

'The trial was designed to achieve adequate power for the primary outcome. '

We have deleted the sentence: "The underlying assumption was also that there was a time limit for including patients in an RCT based both on PCCs' capacity to participate in a complex intervention and maintain adherence to the intervention, as well as a time limit considering changes in society and population, and thus inclusion should not exceed 2.5 years."

as this was not a part of the power analysis, but rather an important factor when calculating how many PCCs that needed to be included to reach the sufficient number of patients during specified time.

5. You refer to some statistically significant results in the subgroup analyses, but there is a large number of statistical tests and it is not clear that multiplicity was accounted for in all the between-arm comparison (at start, 3 months, 6 months, 12 months etc). If multiplicity was not accounted for, the risk of spurious findings due to multiplicity could be acknowledged as a limitation.

ANSWER: We have added: Another limitation was that multiplicity was not accounted for in the analyses of secondary outcomes (page 26).

Reviewer: 2

Prof. Mette Brekke, University of Oslo

Comments to the Author:

Dear authors,

You have answered all my questions and responded to my remarks in a satisfactory way.

The only - very minor - thing I still see as unclear is: Under Methods, study setting: The 'PCCs are publicly or privately driven..' What does 'privately driven' mean? By whom? Certainly publicly funded, not privately?

ANSWER: Thank you for pointing this out. You are perfectly right, all PCCs are publicly funded, so we chose to delete the sentence "The 'PCCs are publicly or privately driven'".